# Soybean Seed Enrichment with Cobalt and Molybdenum as an Alternative to Conventional Seed Treatment

**DOI:** 10.3390/plants12051164

**Published:** 2023-03-03

**Authors:** Cassio Hamilton Abreu-Junior, Gabriel Asa Corrêa Gruberger, Paulo Henrique Silveira Cardoso, Paula Wellen Barbosa Gonçalves, Thiago Assis Rodrigues Nogueira, Gian Franco Capra, Arun Dilipkumar Jani

**Affiliations:** 1Center of Nuclear Energy in Agriculture, Universidade de São Paulo, Av. Centenário 303, Piracicaba 13416-000, Brazil; 2School of Agricultural and Veterinarian Sciences, São Paulo State University, Via de Acesso Prof. Paulo Donato Castellane, s/n, Jaboticabal 14884-900, Brazil; 3Department of Plant Health, Rural Engineering, and Soils, São Paulo State University, Av. Brasil n◦ 56, Ilha Solteira 15385-000, Brazil; 4Dipartimento di Architettura, Design e Urbanistica, Università degli Studi di Sassari, Polo Bionaturalistico, Via Piandanna n◦ 4, 07100 Sassari, Italy; 5Department of Biology and Chemistry, California State University, Monterey Bay, Seaside, CA 93955, USA

**Keywords:** germination, *Glycine max*, plant nutrition, seed quality, Cerrado

## Abstract

Biological nitrogen fixation in soybean is enhanced when seed is treated with cobalt (Co) and molybdenum (Mo) prior to planting. In this study, our objective was to verify if Co and Mo application during the reproductive phase of the crop increases seed Co and Mo concentration without adverse effects on seed quality. Two experiments were conducted. First, we investigated foliar and soil Co and Mo application under greenhouse conditions. Next, we validated the results obtained in the first study. The treatments for both experiments consisted of Co doses combined with Mo, and a control without Co and Mo application. The foliar application was more efficient in producing enriched Co and Mo seed; meanwhile, as Co doses increased so did Co and Mo concentrations in the seed. There were no adverse effects on nutrition, development, quality, and yield of parent plants and seed when these micronutrients were applied. The seed showed higher germination, vigor, and uniformity for the development of soybean seedlings. We concluded that the application of 20 g ha^−1^ Co and 800 g ha^−1^ Mo via foliar application at the reproductive stage of soybean increased germination rate and achieved the best growth and vigor index of enriched seed.

## 1. Introduction

Soybean (*Glycine max* (L.) Merrill) is one of the most important crops in the world, with its grain serving the oil, flour, and beverage production market. [1]. Brazil produces approximately 50% of the global soybean supply [2], and its success can be attributed to research and new technologies conducted in the Cerrado (savanna) region, which has contributed to the expansion of soybean in almost all of Brazil’s territory [3].

Soybean seed protein concentration can range from 27 to 45% [4]. Large amounts of nitrogen (N) are required to maintain this high protein concentration. Research has shown that biological nitrogen fixation (BNF) can supply up to 94% of the total N in soybean. Not only is soybean N demand largely met by BNF, but engagement in BNF reduces the need for mineral fertilizers as well as the negative environmental impacts caused by N fertilizer production [4]. However, for efficient BNF, it is necessary that plant nutrition is adequate, since all nutrients are required to support nodule formation through full functioning of the nitrogenase enzyme [5].

Among the required elements, cobalt (Co) and molybdenum (Mo) have the most relevance in leguminous crops because they are involved in the N cycle, especially in BNF. Cobalt is considered a beneficial element for some groups of plants; however, for legumes it is essential because it acts directly in the formation of nodules and is a component of vitamin B12 (cobalamin), a precursor of leg-hemoglobin, which has the function of preventing the inactivation of the nitrogenase enzyme [6]. Molybdenum is part of two metalloenzymes: nitrogenase, which helps the bacteria in the legume nodules fix atmospheric N, and nitrate reductase, an important enzyme in N assimilation in the plant [7].

However, soil Co and Mo availability is limited by several factors, such as their concentration, soil type and its reaction (pH and CEC), and organic matter concentration [8,9]. The concentration of Co and Mo in the soil has been in decline over the years due to intensive cultivation [10]. Therefore, even though they are required in low quantities, the supply of these elements is necessary, especially during nodulation. Both Co and Mo are normally supplied to soybean by seed treatment (ST), which consists of the direct application of these elements to the surface of seed. However, ST with Co and Mo presents problems to producers, such as the decline in the population of *Bradyrhizobium* in the roots [11]; inhibitory effect on the absorption of iron (Fe) generated by Co [12]; and it also adds a pre-planting step in production. 

Therefore, the acquisition of appropriate technology by producers, the promotion of research, and obtaining new cultivars that are more productive and less susceptible to adverse conditions are important factors for soybean productivity. Among the technologies tested and recommended to soybean producers are the supply and/or enrichment of Co and Mo in the seeds [11,12]. We hypothesize that enriching soybean seed with Co and Mo, via foliar and/or soil application during the reproductive phase of the parent plant, is an alternative to ST for sowing soybean.

The literature does not present a consensus about this practice, especially regarding the viability of enriching the seed with Co, the improvement of the physiological quality of the seed, and the sufficient concentration of these elements in the seed to nourish the new plant. Therefore, the objective of this study was to determine the impact of soil and foliar Co and Mo application during soybean’ reproductive phase on Co and Mo concentrations and the physiological quality of seeds.

## 2. Results

### 2.1. Greenhouse Experiment

The Co concentration in soybean seed and shoots was higher when Co and Mo were applied foliarly than by soil (Table 1). Furthermore, foliar application of Co provided an increase in the concentration of Co in seed and shoots compared to the control without Co and Mo application. Applying Co and Mo by soil increased the Co concentration in the seeds in relation to the control only at the highest dose of Co. There was no difference between the control and the treatments in terms of Co concentration in the roots and soil.

The Co concentration in the seed, shoots, and roots increased as a function of the doses of Co applied foliarly and by soil (Figure 1A,C,D). However, the application of Co by soil did not change its shoot Co concentration. Foliar application provided a larger increase than soil application, ranging from 0.04 to 2.02 and from 0.07 to 0.42 mg kg^−1^ in Co concentration in the seed, respectively. The foliar application of Co increased from 0.5 to 47 mg kg^−1^ in the concentration of Co in the shoots of soybean. In the roots, there was no difference between the forms of application. However, there was an increase from 0.4 to 1.3 mg kg^−1^ in the concentration of Co with the rise in dosage.

There was a higher Mo concentration in the seed with soil application than when foliar application was used; both forms of the application being higher than the control without Co and Mo application (Table 1). The highest Mo concentration in soybean seed was 134 mg kg^−1^ at a dose of 16.3 g ha^−1^ of Co applied via foliar application (Figure 1B). Starting at a dose of 10.8 g ha^−1^ of Co applied by soil, there was an increase in the Mo concentration in soybean, reaching a maximum of 298 mg kg^−1^ in the dose of 30 and 800 g ha^−1^ of Co and Mo (Figure 1B).

The Mo concentration in the soybean shoots was 4.3 times higher under foliar application than by soil, with both applications providing a higher concentration than the control (Table 1). The Mo concentration in the roots and soil were 1.1 and 0.78 mg kg^−1^ when applied foliarly and 76.7 and 3.68 mg kg^−1^ when applied by soil, respectively. Both compartments showed Mo concentrations equal to the control when the foliar application was made.

The forms of Co and Mo application of did not affect the concentration of macronutrients and micronutrients in the soybean seed and shoots (Table 2). Only seed B concentration was affected by increasing doses of Co, with a minimum concentration equal to 18.5 mg kg^−1^, obtained with the application of 14 g ha^−1^ of Co (y = 22.2 − 0.53*x + 0.019*x^2^; R^2^ = 0.65, *p* < 0.05) (data not shown).

### 2.2. Field Experiment

The chlorophyll, flavonoid, and N balance indices, germination rate, and soybean yield were not influenced by Co and Mo application (Table 3). However, applying the highest dose of Co provided a 45% increase in germination rate after accelerated aging compared to the control. In addition, increasing the Co dose provided a 23% linear increase in germination rate after the accelerated aging of soybean seed (Figure 2A).

There was no difference between the control and Co and Mo doses for the growth index (GI) (Table 3). However, the application of 20/800 g ha^−1^ of Co and Mo provided a greater growth index in soybean seedlings concerning the other Co doses (Figure 2B). There was no difference between treatments for the development uniformity index (UI) (Figure 2B). Moreover, applying the highest dose of Co provided higher UI than the control without Co and Mo. For the vigor index (VI), there was a difference between the application of 20/800 g ha^−1^ of Co and Mo and the control (Table 3), and this treatment provided higher seed vigor compared to the others (Figure 2B).

With increasing doses of Co, applied foliarly, there was a linear five- and eight-fold increase in Co concentration in the seeds and the diagnostic leaf of soybeans, respectively (Figure 3A,B). The Mo concentration in soybean did not vary with the increase in the dose of Co, maintaining a concentration of 38 mg kg^−1^, but it was higher than the control, with a concentration of 16 mg kg^−1^. The Mo concentration in the diagnostic leaf was higher with the application of Co compared to the control, with an average concentration of 2.9 mg kg^−1^, and there was an increase up to a dose of 16 g ha^−1^ of Co with a maximum concentration equal to 153 mg kg^−1^. The concentrations of Co and Mo in the soil decreased with the increase in Co doses, and there was no difference between the treatments and the control.

The concentrations of Ca, Mg, S, Fe, Mn, Zn, and Cu in soybean seed were statistically different when treated with Co and Mo (Table 4). Overall, the application: (i) of Co (10–30/800 g ha^−1^ of Co and Mo) decreased the concentration of Ca and Mn; (ii) of Mo (0–30/800 g ha^−1^ of Co and Mo) decreased the concentration of Fe; and (iii) of the highest dose of Co (30/800 g ha^−1^ of Co and Mo) provided a reduction in the concentration of Mg, S, Zn, and Cu, compared to the control. There was no statistical difference between the control and the treatments for the nutrient concentration in the diagnostic soybean leaf, except for B, where the application of only Mo (0/800 g ha^−1^ of Co and Mo) increased the concentration of B. The application of Co provided a linear reduction of 14, 12, 15, 14, 13, and 29% of the concentration of Ca, Mg, S, Cu, Mn, and Zn in the seeds and a linear reduction of 21 and 24% of the concentration of B and Mn in the diagnostic leaf of soybean, respectively.

Cobalt concentration in soybean seed was positively correlated (r = 0.46–0.90; *p* < 0.05) to the shoot and roots Co concentration and with seed and shoot Mo concentration in the parent plants (Table 5). Molybdenum concentration in soybean seed was positively correlated (r = 0.46–0.90; *p* < 0.05) with shoot Co and Mo concentration in the parent plants.

## 3. Discussion

In soybean, approximately 40% of the grain reserves are proteinaceous, which demands high amounts of N, and in Brazil, most of the N in soybean crops comes from BNF [4]. Therefore, the supply of Co and Mo should be performed at the early stage of legume cultivation since the availability of these elements can be affected by biological [13], physical, and chemical properties of the soil, such as soil organic matter and pH [14]. Thus, enriching soybean seed with these elements can ensure BNF efficiency. Co is required for regulating O_2_ input into the nitrogenase enzyme complex [7]. Mo, in turn, tends to increase the protein concentration of soybean crops [15] because it is directly involved in the N cycle, being (i) influential in the enzyme reductase (such as nitrate), (ii) part of the reaction center of the nitrogenase enzyme complex, and (iii) a component of the Fe-Mo protein (NifDK) [16,17].

We observed that the concentration of Co in soybean seed and leaves was higher when the element was applied foliarly than by soil (Table 1) and increased linearly with the increase in Co doses (Figure 1). This result highlights the better redistribution of Co in the parent plants when applied foliarly to produce enriched seeds with Co.

We observed the opposite for Mo, where soil application provided higher levels of Mo in the seed than foliar application (Table 1), showing the better movement of the element in the plant through the process of distribution rather than redistribution. The Mo levels in the leaves were also higher when there was foliar application, which may have occurred due to both the low redistribution of Mo and the presence of residues from applying the element directly to the leaves. However, although the dose of Mo was constant among treatments, soil application of increasing doses of Co provided higher seed Mo concentration, while foliar application obtained maximum Mo concentrations in seed at the dose of 16 g ha^−1^ of Co (Figure 1B). Furthermore, the foliar application of Co chelates and sodium molybdate did not affect the biometric characteristics of the plant under controlled conditions (Appendix A).

We observed similar results in the experiment under field conditions, as the foliar application of Co doses increased the Co concentration in soybean seed and leaves (Figure 3). The Mo concentration in soybean leaves increased up to the 16 g ha^−1^ Co application rate, even though the application of Mo occurred at the same dose (800 g ha^−1^). Despite the increase in Mo concentration in the leaves, the Mo concentration in soybean was not altered (Figure 3D), again showing the low mobility of this element in the plant.

Foliar application of Co and Mo provided a 24% higher concentration of Co in the seed compared to soil application. The mobility of Co in phloem is considered intermediate [18]. However, foliar application at the R5 (beginning of grain filling) stage increased the levels of this element in developing organs (Table 1). The Mo remobilization capacity was low; the nutrient applied to tissue may not be remobilized to developing tissue. Consequently, the application via soil is more efficient since the element will be redirected to the site of greatest activity in the plant, i.e., the younger tissues.

However, the feasibility of producing Mo-enriched seed was demonstrated through two foliar applications between R3 (beginning of pod formation) and R5 stages, with a total application of 800 g ha^−1^. Such seeds, in a field trial on soil with low N availability, produced plants with higher N and Mo concentration in the grain and higher yields. Furthermore, those Mo-enriched seeds did not require additional Mo fertilization. In our study, foliar application of Mo provided seeds with Mo concentrations similar to or higher than that observed in this previous research [19].

The uptake of Co and Mo by plant roots is influenced by numerous factors, such as organic matter concentration and the concentration of other elements in the soil, clay, and pH [9], the latter being particularly important. Lowering soil pH tends to increase Co uptake by roots by increasing the phytoavailability of this metal [20]. Cobalt is taken up as Co^2+^ via the IRT1 transporter, the same transporter as Fe and other cationic metals in *Arabidopsis thaliana* [21]. With this, Co uptake may be compromised in soils with low Co concentrations due to competition with other metal elements. The amount of Co that a plant takes up depends on its Fe status; as Fe deficiency increases, the higher the expression of IRT1, thus more Co is taken up [22]. We observed it in the field experiment, that increasing the Fe concentration in the plant decreased the Co concentration in soybean seeds (r = −0.53, *p* < 0.01; Appendix A).

To reduce costs with N fertilization in leguminous plants, such as soybean, the efficiency of BNF should be guaranteed based on an adequate nutritional supply to the plant [4]. In the process of the symbiosis of some microorganisms with legumes, there is an increase in demand for elements such as B, Ca, Co, Cu, Cu, Fe, K, Mo, Ni, P, Se, and Zn, which play specific functions in plant metabolism [5]. Thus, the enrichment of soybean seed with Co and Mo is essential for full plant development and efficient performance of BNF when those seeds are seeded in dystrophic soils. Nevertheless, the interaction of these elements with the other nutrients the plant and seeds require should be better evaluated.

In our greenhouse experiment, the seed B concentration increased with increasing Co doses. However, the opposite was observed in the field experiment, where the application of Co and Mo reduced the concentration of B and Mn in the leaves and of Ca, Cu, Mg, Mn, S, and Zn in soybean seeds. This seems related to the different experimental conditions between greenhouse vs. field, such as the soil properties and the soil volume explored by the roots. Additional reasons could be related to evapotranspiration and the different phases in which shoots were sampled. Despite these results, the levels of B, Ca, Fe, Mg, Mn, P, S, and Zn in soybean leaves, for both experiments, were within an adequate range for soybean crops, which is 2.5–5.0, 4–20, 3.0–10.0, 2.1–4.0, 21–55, 50–350, 20–100, and 20–50 mg kg^−1^ [23], respectively; while N, K, and Cu concentrations in the leaves were 22, 15, and 3.2 mg kg^−1^ in the greenhouse experiment and 34, 9.5, and 5.4 mg kg^−1^ in field conditions, which were below the optimal range of 40–54, 17–25, and 10–30 mg kg^−1^ [23], respectively, being then considered in nutritional deficiency. However, leaf sampling occurred between stages R6 (grain filled) in the field and R7 (beginning of grain maturation) in the greenhouse. In contrast, sampling at stage R2 (full flowering) was recommended [22], and these elements may have already been redistributed to the seed.

Cobalt has beneficial effects on leguminous plants; however, there is a narrow line between plant need and toxicity [24]. High doses can cause oxidative stress in plants [25]. Under field conditions, the physiological and yield characteristics of the plant were not influenced by Co and Mo application at the reproductive stage of the crop. The seed were tested in the laboratory to assess their germination quality. We observed that the doses of Co and Mo application linearly increased the seed germination rate after accelerated aging (Figure 2A) and increased the seed vigor index and seedling growth index with the doses of 20 g ha^−1^ of Co (Figure 2B). Seeds with higher Co concentrations trigger endogenous ethylene synthesis, leading to higher germination rates and seedling development [26].

Additional research is warranted to investigate the BNF efficiency and soybean yield when seed is enriched with Co and Mo. Comparing crops derived from seed treated with traditional Co and Mo application methods, such as seed treatment and soil and/or foliar application, will be the next step in our research.

## 4. Materials and Methods

### 4.1. Experimental Design

Two experiments were conducted. First, under greenhouse conditions, we investigated foliar or soil application of Co and Mo. In the second one, under field conditions, we sought to validate the results obtained under greenhouse conditions.

#### 4.1.1. Greenhouse Experiment

The first experiment was conducted in a greenhouse located at the Center for Nuclear Energy in Agriculture (CENA), of the University of São Paulo (USP), at the geographic coordinates 22°42′30″ S and 47°38′00″ W, at 524 m of altitude, in the State of São Paulo, Brazil.

Plastic pots with 3 dm³ capacity, coated with plastic bags, and filled with 3.2 kg of soil samples of a Typic Quartzipisamment [27], collected from the 0 to 20 cm layer, were used. The physical and chemical properties of the soil are shown at Table 6. Soil pH was measured potentiometrically with a glass electrode at the soil/solution mixture of 1:2.5 1 mol L^−1^ CaCl_2_. Soil organic matter (SOM) content was estimated by the Walkley–Black method. Soil labile P, Ca^2+^, K^+^, and Mg^2+^ were extracted by an ion-exchange resin procedure. Concentrations of P in the extracts was determined by the colorimetric method, and of Ca and Mg by atomic absorption spectrophotometer (AAS) using a VARIAN SpectrAA 140 and K using a CORNING 400 flame-photometer. Total acidity (H + Al) was determined by SMP buffer solution at pH 7.0. (H + Al) = Acidity at pH = 7.0, determined by SMP buffer solution at pH 7.0. Al^3+^ extracted with 1.0 mol L^−1^ NH_4_Cl. CEC = Cation exchange capacity (Ca^2+^ + K^+^ + Mg^2+^ + (H + Al)). Cu, Fe, Zn, and Mn were extracted by DTPA solution at pH 7.3 while B determined by the hot water method (samples sieved at 0.5 mm) [28]. The densimeter method was used for particle-size analysis [29]. To raise the soil base saturation to 60% and the magnesium (Mg) concentration to 5 mmol_c_ dm^−3^, CaCO_3_ + MgCO_3_, in the stoichiometric relation 4:1, was added in each pot at a dose corresponding to 0.5 t ha^−1^. The soil samples and the correctives were homogenized and incubated, being watered with deionized water to maintain soil moisture 60% water holding capacity, for 45 days, without drainage.

##### Seed Treatment and Cultivation Conditions

The soybean cultivar used was FTS1154RR (FTS), non-commercial, super early cycle (95 days in field conditions), belonging to maturity group 5.1 (beginning of grain filling), with an indeterminate growth habit. The seed was previously treated with *Bradyrhizobium japonicum* liquid inoculant, 10 mL kg^−1^ of seeds. A mixture of the SEMIA 5079 and SEMIA 5080 strains was used, with 6 × 10^9^ colony forming units per milliliter. Sowing was performed manually, distributing five seed per pot and, after 15 days, thinning was performed, keeping two plants per pot. At the same time as sowing, maintenance fertilization was performed applying the following doses, in mg kg^−1^ of soil: 200 of P; 150 of K; 75 of Ca; 15 of Mg; 50 of S; 0.5 of B; 1.5 of Cu; 5 of Fe; and 5 of Zn per pot [30]. The soil was moistened with deionized water to approximately 70% of water holding capacity, once a day, until plants were harvested at developmental stage R7 (beginning of grain maturation).

##### Application of Co and Mo

Cobalt and Mo applications, according to the treatments, were made when the plants reached the R5.4 development stage (most pods in the upper third of the main stem with 51 to 75% of maximum granulation). Cobalt and Mo sources were Co chelate and sodium molybdate with 13.7 and 39% (*w*/*w*) Co and Mo, respectively. Five percent of mineral oil was added to the application solution. Foliar application was performed using a 4 cm flat brush and precision balance, in order to ensure the correct application dose and to avoid solution runoff from the leaf to the soil. The soil application of the Co and Mo solutions was performed directly on the soil surface.

The experiment was set up In a co”plet’ly randomized design with a 4 × 2 + 1 factorial scheme and three repetitions. The factors consisted of Co application in four doses: 0, 10, 20, and 30 g ha^−1^, together with application of 800 g ha^−1^ of Mo, in two forms of application: soil and foliar. The additional treatment consisted of soybean cultivation without the application of Co and Mo.

##### Evaluation of Production Components

At developmental stage R7, the plants were sectioned at the height of the soil surface and separated into roots, shoots, and seeds, and the soil was sampled. The roots, in a sieve, were separated from the soil by washing in running water. The roots and shoots were washed with deionized water. Afterwards, together with the seeds, they were dried in a forced air circulation oven at 60 °C for 72 h. Afterwards, the dry mater of the shoots, roots, and seeds were weighed and ground in a Wiley stainless steel mill and stored in a plastic bag until the nutrients (N, P, K, Ca, Mg, S, B, Cu, Fe, Mn, Mo, and Zn) and Co analysis.

#### 4.1.2. Field Experiment

##### Installation and Treatments of the Experiment

The second experiment was conducted under field conditions at the Geraldo Schultz Research Center, municipality of Iracemápolis (22°38′46.2″ S and 47°30′11″ W, at 530 m of altitude), in São Paulo State, Brazil. In this period, the total precipitation was 830 mm and average maximum and minimum temperatures of 29 °C and 19 °C, respectively. The experimental area has an Oxisol [27] (Table 1).

The treatments consisted of the Co application in doses of 0, 10, 20, and 30 g ha^−1^, together with 800 g ha^−1^ of Mo, both via foliar application. There was also an additional treatment without the application of Co and Mo. The experiment was set up in a randomized complete block design with six replications. The plots had an area of 18 m² with six rows of 6 m long and 15.5 plants per linear meter, with spacing of 0.5 m between rows, corresponding to 350,000 plants per hectare. The central 5 m of the two central rows of each plot was considered as usable area.

The soybean cultivar was M6410IPRO, medium cycle (approximately 120 days under field conditions), belonging to maturity group 6.4, with indeterminate growth habit. The treatment of the seeds for planting was the same as that described for the experiment in the greenhouse. The seeds also received a fungicide and insecticide application at a dose of 0.1 L ha^−1^, commercial product (active ingredient concentration, 25 g L^−1^ of Pyraclostrobin, 225 g L^−1^ Thiophanate Methyl, and 225 g L^−1^ Fipronil, 250 g L^−1^). Sowing was performed using a tractor-driven fertilizer drill. Base fertilization consisted of the application of 200 kg ha^−1^ of NPK formula 09-48-00, followed by pre-planting fertilization of 200 kg ha^−1^ of KCl. The cultural treatments applied to the experiment consisted of two applications of glyphosate and four applications of fungicides and insecticides.

The application of Co and Mo, according to the treatments, occurred when the plants reached the R5.4 stage, when the crop had a closed canopy, reducing the runoff of the solution into the soil. A CO_2_ pressurized portable knapsack sprayer was used, keeping the spray tips at a height of approximately 0.50 m from the top of the soybean crop, at a working pressure of approximately 25 psi, with a spray volume of 150 L ha^−1^.

##### Assessment of Production Components

When the plants reached the R.6 stage (full pod filling), the chlorophyll (CI), flavonoid (FI), and N balance indexes (NBI) were evaluated on the central trifoliate of the third leaf in three plants per plots, using DUALEX DX4 equipment. At the same time, leaf sampling was performed for leaf diagnosis, collecting the third leaf from the apex of the plants, collecting 30 leaves per plot.

At 127 days after planting, the plants of the useful area were manually cut close to the ground and then sent to a stationary grain thresher to obtain the seed. The manual processing of the grain was performed to remove impurities, burnt, wrinkled, damaged and contaminated grain by primary infection. At harvest, grain moisture was reported at 13%. Drying and milling of the samples was the same as with the samples in the greenhouse.

At harvest (127 days after planting), soil was sampled at 0–20 cm depth from six points to make up one composite sample per plot to determine Co and Mo concentration.

#### 4.1.3. Germination Test

To estimate the seed germination rate (GR), germination paper moistened with an amount of water equivalent to 2.5 times the mass of dry paper used and 50 seeds from each plot was used. Subsequently, the seed was taken to a germination chamber at a constant temperature of 25 ± 1 °C. The evaluation was performed at 4 and 7 days after sowing [31].

#### 4.1.4. Accelerated Aging Test

For the accelerated aging test (AAGR) [32], four samples of 50 seeds were distributed in a single layer on a stainless-steel screen and placed inside plastic boxes of the “gerbox” type, containing 40 mL of distilled water, with approximately 2 cm of distance between the water level and the seeds. Then, the boxes were closed and taken to a DBO-germination chamber, regulated at a temperature of 42 ± 1 °C, for 48 h. After the aging period, the seeds were evaluated to the germination rate (AAGR), as described above. The evaluation was performed on the fourth day after transfer to the germinator.

#### 4.1.5. Computerized Analysis of Seedlings Vigor with SVIS^®^ Software 

To obtain seedlings for evaluation, five replications of 20 seeds were placed to germinate in two rows positioned in the upper third of the surface of germination paper, at 25 °C, for three days [33,34]. After the germination period, the seedlings were transferred to a black cardboard sheet and the images were captured in an HP Scanjet 2004 scanner, mounted in an inverted manner inside an aluminum box measuring 60 × 50 × 12 cm and operated by Photosmart software, with 100 dpi resolution. The scanned images were analyzed by Seed Vigor Imaging System (SVIS^®^) software.

The program generated the growth index (GI) and the development uniformity index (UI) of seedlings. The data were obtained considering 12.7 cm (5 inches) as the maximum seedling size to be reached within three days. For composition of the vigor index (VI), the following combination was used: VI (70/30) = (0.7 × GI) + (0.3 × UI) [34]. The value of the vigor index results from the combination of growth and uniformity indices, which can range from 0 to 1000.

### 4.2. Chemical Analysis

The soil samples were air dried and passed through a sieve with a mesh aperture of 1.25 mm. Subsequently, for determination of the Co and Mo concentrations, an acid digestion (HNO_3_ + H_2_O_2_ + HCl) [35]. The plant samples (roots, aerial part, and seeds) were digested with a nitric-perchloric acid in a 3:1 ratio, in a digester block system [36]. In the extracts that were obtained, the macronutrients (P, K, Ca, Mg, and S), micronutrients (B, Cu, Fe, Mn, Mo, and Zn), and Co concentrations were determined using an inductively coupled plasma optical atomic emission spectrometer (ICP-OES). The N concentration was determined by the Kjeldahl method [36].

### 4.3. Statistical Analysis

The data from each experiment were analyzed separately and according to the experimental design. The data were submitted to ANOVA by the F test (*p* < 0.05). If the treatments were significant, the additional treatment was compared with the treatments by the Dunnett test (*p* < 0.05), multiple comparison of means by the Tukey test (*p* < 0.05), and regression equations were adjusted, testing the significance of the coefficients by the *t* test (*p* < 0.1). The concentration of Co and Mo in plant parts and in soil were correlated by Pearson’s test (*p* < 0.05). RStudio version 3.4.1 [37] was used for statistical analysis.

## 5. Conclusions

Foliar application of Co and Mo at the R5.4 reproductive stage of soybean parent plants increases the concentration of Co and Mo in seed. The application of Co and Mo did not negatively impact soybean nutrition, development, quality, as well as the yield of both parent plants and seed.

Under field conditions, the doses of Co increase the germination rate after accelerated aging and achieve the best growth index and vigor index of generated enriched Co (0.8 mg kg^−1^) and Mo (38 mg kg^−1^) seeds at dose of 20 g ha^−1^ of Co, plus 800 g ha^−1^ of Mo, applied via foliar at the reproductive stage of soybean parent plants.

## Figures and Tables

**Figure 1 plants-12-01164-f001:**
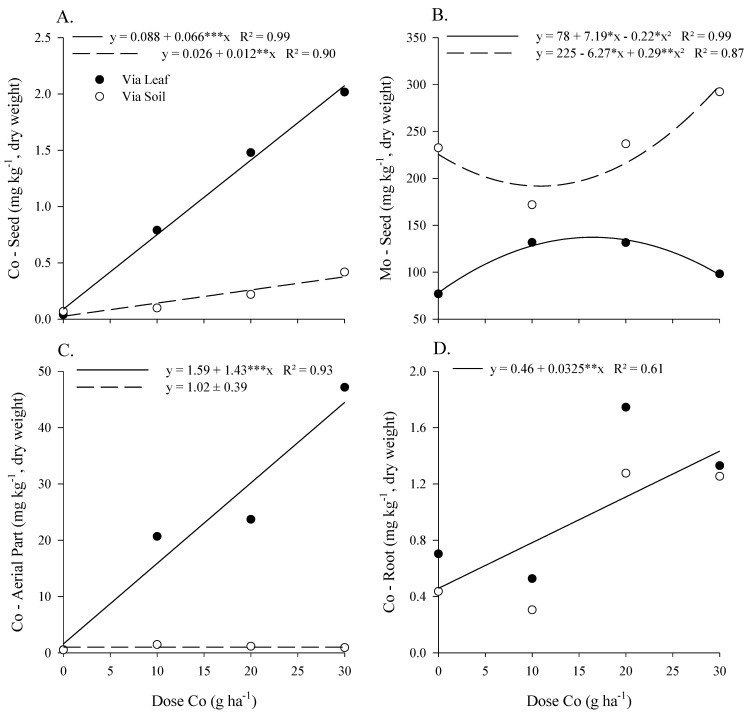
Cobalt and Mo concentration in seed (**A**,**B**) and Co in aerial part (**C**) and soybean roots (**D**) as a function of Co doses, with 800 g ha^−1^ Mo, via leaf and via soil application in soybean parent plants, in the maturation phase. *, **, ***-Significant at 0.05, 0.01, and 0.001 by *t*-test.

**Figure 2 plants-12-01164-f002:**
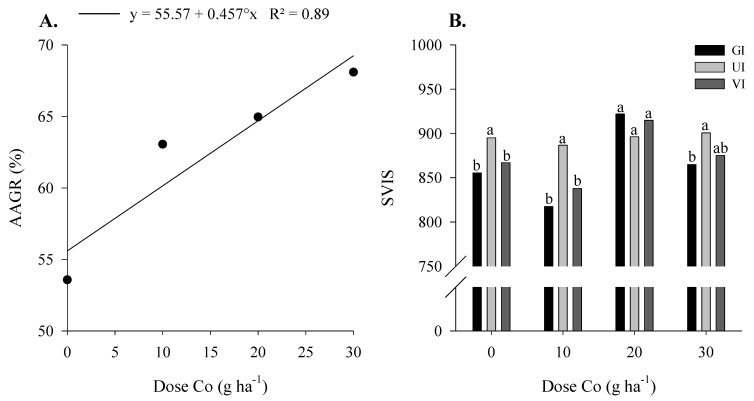
Accelerated aging germination rate (AAGR) (**A**), growth index (GI), growth uniformity index (UI), and vigor index (VI) (**B**) of soybean seeds as a function of foliar Co and Mo doses, applied to parent plants foliarly, in the maturation phase, under field conditions. °-Significant by *t* test (*p* < 0.10). Bars followed by the same letter do not differ by Tukey test (*p* < 0.05).

**Figure 3 plants-12-01164-f003:**
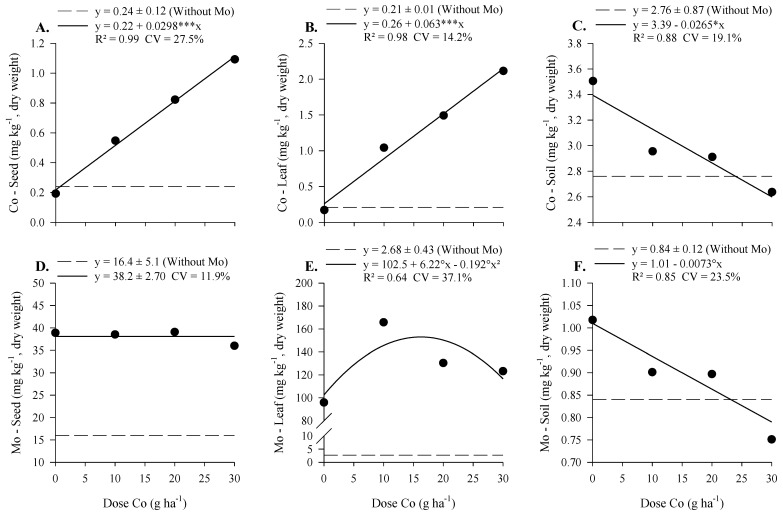
Cobalt and Mo concentration in seed (**A**,**D**), in diagnosis leaf of soybean (**B**,**E**), and soil (**C**,**F**) as a function of Co doses and Mo applied to parent plants via foliar in the maturation phase, under field conditions. °, *, and ***—Significant at 10, 5, and 0.1% probability by *t*-test.

**Table 1 plants-12-01164-t001:** Cobalt and Mo concentrations in seeds, shoots, roots, and soil as a function of foliar and soil Co and Mo application in soybean parent plants during maturation.

Dose (g ha^−1^)	Foliar	Soil	Foliar	Soil	Foliar	Soil	Foliar	Soil
	Seeds		Shoots		Roots		Soil	
Co/Mo	--------------------- Concentration of Co (mg kg^−1^, dry weight) ---------------------							
0/0	0.002 B		0.40 B		0.69 A		0.19 A	
0/800	0.04 aB	0.07 aB	0.5 aB	0.5 aB	0.7 A	0.4 A	0.20 A	0.25 A
10/800	0.79 aA	0.10 bB	20.7 aA	1.5 bB	0.5 A	0.3 A	0.19 A	0.27 A
20/800	1.48 aA	0.22 bB	23.7 aA	1.2 bB	1.7 A	1.3 A	0.18 A	0.25 A
30/800	2.02 aA	0.42 bA	47.2 aA	0.9 bB	1.3 A	1.3 A	0.21 A	0.22 A
Mean	1.08	0.20	23.0	1.0	1.1 a	0.8 a	0.20 b	0.25 a
CV (%)	19.9		76.4		54.0		23.2	
Co/Mo	--------------------- Concentration of Mo (mg kg^−1^, dry weight) ---------------------							
0/0	48 B		10.6 B		0.10 B		0.52 B	
0/800	77 bB	232 aA	1116 A	194 A	0.0 B	63 A	0.82 B	3.59 A
10/800	131 bA	172 aA	1209 A	250 A	1.4 B	56 A	0.70 B	3.84 A
20/800	132 bA	237 aA	821 A	287 A	2.2 B	77 A	0.90 B	4.04 A
30/800	98 bB	292 aA	1179 A	283 A	0.7 B	110 A	0.70 B	3.25 A
Mean	110	233	1081 a	253 b	1.1 b	76.7 a	0.78 b	3.68 a
CV (%)	19.1		44.9		71.5		17.1	

Means followed by the same capital letter (column) do not differ from the control (Dunnett test; *p* < 0.05). Means followed by the same lower-case letter (line) do not statistically differ (Tukey test; *p* < 0.05).

**Table 2 plants-12-01164-t002:** Macro- and micronutrient concentrations in parts of soybean parent plants under foliar and soil Co and Mo application application, in the maturation phase, under controlled conditions.

Plant Part	Application Forms	N	P	K	Ca	Mg	S	B	Cu	Fe	Mn	Zn
		---------------------- g kg^−1^, Dry Weight -------------------						--------------- mg kg^−1^, Dry Weight --------------				
Seeds	Via Foliar	60 a	8.2 a	19.9 a	1.9 a	2.3 a	3.4 a	22.3 a	3.6 a	51 a	34 a	42 a
	Via Soil	57 a	8.0 a	18.4 a	1.8 a	2.2 a	3.3 a	19.4 a	3.3 a	50 a	37 a	40 a
	CV (%)	5.9	15.1	12.6	31.6	12.0	13.8	17.7	20.8	16.8	15.3	16.8
Shoots	Via Foliar	21 a	11.5 a	15.3 a	10.5 a	1.6 a	2.6 a	53 a	3.0 a	174 a	41 a	24 a
	Via Soil	23 a	10.7 a	15.6 a	9.4 a	1.8 a	2.3 a	61 a	3.5 a	218 a	53 a	29 a
	CV (%)	9.1	19.7	26.5	15.0	13.0	48.7	33.9	52.1	28.8	35.1	47.3
Roots	Via Foliar	-	0.85 a	1.56 a	2.6 a	1.1 a	2.6 a	40.7 b	11.3 a	3521 a	52 a	40 a
	Via Soil	-	1.19 a	2.09 a	3.0 a	2.1 a	3.5 a	73.2 a	12.9 a	3370 a	81 a	44 a
	CV (%)	-	88.7	69.4	56.4	50.8	39.8	23.4	22.6	34.2	49.1	37.2

Means followed by the same letter (column) do not statistically differ (Tukey test; *p* < 0.05).

**Table 3 plants-12-01164-t003:** Chlorophyll (CI), flavonoid (FI), and nitrogen balance indexes (BNI) of leaves, seed yield and germination rate (GR), accelerated aging germination rate (AAGR), growth index (GI), development uniformity index (UI), and vigor index (VI) of soybean seeds as a function of Co and Mo doses, applied to parent plants foliarly in the maturation phase, under field conditions.

Dose (g ha^−1^)	CI	FI	BNI	Yield	GR	AAGR	GI	UI	VI
Co/Mo	----------- Dualex -----------			kg ha^−1^	------- % -------		-------- SVIS^®^ ----------		
0/0	36 A	0.63 A	58 A	4448 A	93 A	47 B	834 A	876 B	846 B
0/800	34 A	0.60 A	57 A	4190 A	91 A	54 B	855 A	895 B	867 B
10/800	35 A	0.57 A	61 A	4186 A	92 A	63 B	817 A	887 B	838 B
20/800	35 A	0.64 A	55 A	4298 A	92 A	65 B	922 A	896 B	915 A
30/800	35 A	0.61 A	59 A	4588 A	94 A	68 A	865 A	901 A	875 B
CV (%)	6.4	12.9	13.2	17.8	3.4	22.7	3.8	1.6	2.8

Means of the treatments followed by the same letter do not differ from the control (0/0) by the Dunnett test (*p* < 0.05).

**Table 4 plants-12-01164-t004:** Macro and micronutrient concentration in soybean seeds and shoots as a function of foliar Co and Mo doses applied to parent plants foliarly in the maturation phase, under field conditions.

Dose (g ha^−1^)	N	P	K	Ca	Mg	S	B	Cu	Fe	Mn	Zn
	------------------------ g kg^−1^, Dry Weight ------------------------						--------------- mg kg^−1^, Dry Weight ---------------				
Co/Mo	----------------------------------------------------- Seeds -----------------------------------------------------										
0/0	57 A	5.1 A	19.1 A	2.5 A	3.1 A	3.9 A	40 A	12.9 A	90 A	36 A	52 A
0/800	56 A	5.0 A	19.5 A	2.3 A	3.1 A	3.9 A	41 A	12.3 A	72 B	35 A	45 A
10/800	55 A	4.7 A	17.8 A	2.1 B	2.9 A	3.6 A	38 A	12.4 A	64 B	32 B	40 A
20/800	57 A	4.7 A	19.5 A	2.2 B	3.0 A	3.8 A	42 A	11.0 A	64 B	32 B	40 A
30/800	56 A	4.7 A	18.4 A	1.9 B	2.7 B	3.3 B	35 A	8.6 B	64 B	30 B	39 B
L	^ns^	^ns^	^ns^	***	***	**	^ns^	^ns^	***	**	**
Q	^ns^	^ns^	^ns^	^ns^	^ns^	^ns^	^ns^	^ns^	^ns^	^ns^	^ns^
CV (%)	5.6	10.2	11.5	6.0	3.8	7.7	7.8	9.3	4.2	6.5	18.0
Co/Mo	-------------------------------------------------- Shoots --------------------------------------------------										
0/0	36 A	2.9 A	8.6 A	19.0 A	5.6 A	2.6 A	46 B	5.1 A	82 A	173 A	20 A
0/800	31 A	2.7 A	9.7 A	19.6 A	5.5 A	2.6 A	55 A	5.3 A	81 A	167 A	22 A
10/800	34 A	2.7 A	9.9 A	17.7 A	5.2 A	2.6 A	50 B	6.3 A	82 A	152 A	26 A
20/800	36 A	2.3 A	9.4 A	18.0 A	5.8 A	2.4 A	49 B	5.3 A	78 A	153 A	26 A
30/800	36 A	2.2 A	10.0 A	17.8 A	4.6 A	2.5 A	42 B	5.0 A	84 A	122 A	27 A
L	^ns^	^ns^	^ns^	^ns^	^ns^	^ns^	**	^ns^	°	^ns^	^ns^
Q	^ns^	^ns^	^ns^	^ns^	^ns^	^ns^	^ns^	^ns^	^ns^	^ns^	^ns^
CV (%)	16.9	29.9	26.4	12.4	24.5	25.5	11.5	15.1	24.6	32.2	24.0

Means followed by the same letter, vertically, do not differ from the control by the Dunnett test (*p* < 0.05). °, **, and ***-Significant 0.10, 0.01, and 0.001% probability by *t*-test. L = Linear regression. Q = Quadratic regression.

**Table 5 plants-12-01164-t005:** Pearson’s correlation between Co and Mo concentrations in seeds, shoots, and roots of soybean under controlled and field conditions.

Experiment	Attributes		Co				Mo		
			Seed	Shoot	Root	Soil	Seed	Shoot	Root
Greenhouse	Co	Seed	-	-	-	-	-	-	-
		Shoot	0.84	-	-	-	-	-	-
		Root	0.63	0.42	-	-	-	-	-
		Soil	0.15	0.24	0.13	-	-	-	-
	Mo	Seed	0.59	0.46	0.32	−0.06	-	-	-
		Shoot	0.51	0.69	0.28	0.13	0.56	-	-
		Root	0.33	0.16	0.42	0.18	0.05	0.25	-
		Soil	0.34	0.04	0.35	−0.04	0.49	0.16	0.19
Field	Co	Seed	-	-	-	-	-	-	-
		Shoot	0.90	-	-	-	-	-	-
		Soil	−0.23	−0.29	-	-	-	-	-
	Mo	Seed	0.48	0.43	-	0.16	-	-	-
		Shoot	0.46	0.55	-	−0.18	0.75	-	-
		Soil	−0.27	−0.32	-	0.44	0.17	−0.16	-

Red, orange, and yellow filled cells show that there was correlation at 0.05, 0.01, and 0.001 by the *t*-test, respectively. Without filling, there was no correlation at 5% probability by the *t*-test.

**Table 6 plants-12-01164-t006:** Chemical ^(1)^ and physical ^(2)^ properties of soils used for the experiments in greenhouse and in field, respectively.

Physical-Chemical Properties	Unit (in Terms of Soil Mass or Volume)	Greenhouse	Field
pH (CaCl_2_)	-	4.8	5.3
Resin-P	mg kg^−1^	9.0	21
K^+^	cmol_c_ dm^−3^	0.3	2.8
Ca^2+^	cmol_c_ dm^−3^	20	66
Mg^2+^	cmol_c_ dm^−3^	2.0	25
Al^3+^	cmol_c_ dm^−3^	0.0	1.0
H^+^ + Al^3+^	cmol_c_ dm^−3^	25	34
CEC	cmol_c_ dm^−3^	49	127
SO_4_^2−^	mg kg^−1^	46	42
B	mg kg^−1^	0.43	2.3
Cu	mg kg^−1^	0.20	0.30
Fe	mg kg^−1^	18.0	2.70
Mn	mg kg^−1^	0.40	5.0
Zn	mg kg^−1^	0.20	33
Co	mg kg^−1^	0.20	3.10
Mo	mg kg^−1^	0.50	0.80
Clay	%	9.90	56.0
Silt	%	0.10	13.0
Sand	%	90.0	31.0

Soil analysis according to the analytical methods recommended by ^(1)^ Raij [28] and ^(2)^ Camargo [29], respectively. CEC = cation-exchange capacity; BS = base saturation.

## Data Availability

https://teses.usp.br/teses/disponiveis/64/64133/tde-19052017-144600/en.php (accessed on 19 January 2023).

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
