# Peer review of "Soybean Seed Enrichment with Cobalt and Molybdenum as an Alternative to Conventional Seed Treatment"

_plants, 2023, doi:10.3390/plants12051164_

Round 1

Reviewer 1 Report

Dear Authors,

The manuscript plant-2204069 titled “Soybean seed enrichment with cobalt and molybdenum as an alternative to conventional seed treatment” was interesting. Authors firstly got soybean seeds with high cobalt and molybdenum concentration in the greenhouse experiment, and then check seed quality of parent plants via field experiment. However, I still have some minor comment for improving the quality of this manuscript. I still suggest authors to ask native English speakers for checking the English.

Minor comments

Line 34, add the full name for Co and Mo when they were firstly appeared in the Abstract section.

Line 46, add the name after g kg-1.

Please just use the abbreviation of cabalt and molybdenum as Co and Mo after line 54, as you have mentioned their abbreviation in line 54.

Line 62, it is better to point out which elements for the firstly appearance in this paragraph.

Line 86, Co in, but not Coin.

Line 91, it is should be Figure A, Figure C, Figure D.

Line 92, I suggest authors change the element to Co.

In the Table 1, what is the unit after mg kg-1 for the concentrations of Co or Mo, is it mg kg-1 dry weight?

Line 121-124, I didn’t find figure or table for the formula, please add this information.

Line 132-134, change to g ha-1, and change t test to t-test, and in other tables and figures. Is it Co concentration in shoots (C)? Should add the unit of Figure A-D as mg kg-1 dry matter, and also in other place of this manuscript. I thought you measured the concentrations using dried samples.

Line 121-124, please add the source of this sentence, Figure 1?

Line 143, move (BNI) after the ‘indexes’.

The full name of AAGR should be unified in Table 3 and Figure 2, and in the text, and change the abbreviation of IC, IU, IV in Figure 2B to GI, UI, and VI.

Line 139, and Line 142, it should be Figure 2B.

Line 311, in Table 6, change and Raji[28] Camargo[29] to ‘Raji[28] and Camargo[29]’. In this Table, is the unit is mg kg-1 soil, I suggest authors to add soil to make clearer, and similar line 322.

Line 327, use abbreviation for Co and Mo in materials and methods, as you have started used them from line 54.

Line 330, what is the unit of 39%, is it w/v, or w/w?

I couldn’t clearly understand the meaning of R3, R5, R5.4, could you add more information for these abbreviations? Please move (beginning of grain maturation) in line 341 to line 325, after the ‘R7’.

Line 347, authors also measured Mo concentration, I suggest authors to add Mo analysis as well in this sentence.

Line 379, move ‘(NBI)’ after ‘indexes’.

Line 280-Line 289, Line 406-Line 413, change the font following the instruction of Plants-Basel.

Line 452, add (DM) after dry matter.

Table S2, I suggest authors to separate Co and Mo using a line between parameters of soil (Co) and seed (Mo).

Author Response

Responses to reviewer 1

Dear reviewer,

The authors would like to sincerely thank you for the useful comments and suggestions, which were carefully considered and incorporated into the revised manuscript. As you have suggested in your comments, we have asked a native English speaker to check the English. This was performed by Prof. Dr. Arun Dilipkumar Jani, who is a professor at California State University, Monterey Bay, Seaside, CA, USA. The revised manuscript was also improved: 1) the introduction was improved to provide sufficient background. Relevant references were also included, 2) cited references were contextualized to highlight the relevance of the research, 3) the research design was improved, 4) the description of methods was improved, 5) the conclusions were improved to be supported by the results.

Minor comments

Line 34, add the full name for Co and Mo when they were firstly appeared in the Abstract section.

R.: Done.

Line 46, add the name after g kg-1.

R.: Done.

Please just use the abbreviation of cabalt and molybdenum as Co and Mo after line 54, as you have mentioned their abbreviation in line 54.

R.: Done. However, note that when beginning a sentence, abbreviations should not be used. Thus, we spelled out Co and Mo when they were the first words in sentences.

Line 62, it is better to point out which elements for the firstly appearance in this paragraph.

R.: Done.

Line 86, Co in, but not Coin.

R.: Done.

Line 91, it is should be Figure A, Figure C, Figure D.

R.: Done.

Line 92, I suggest authors change the element to Co.

R.: Done.

In the Table 1, what is the unit after mg kg-1 for the concentrations of Co or Mo, is it mg kg-1 dry weight?

R.: Done.

Line 121-124, I didn’t find figure or table for the formula, please add this information.

R.: Yes, the formula is not presented in any table or figure. And it is only presented for boron (B) because it was for the unique micronutrient affect by Co doses and data were nor shown.

Line 132-134, change to g ha-1, and change t test to t-test, and in other tables and figures. Is it Co concentration in shoots (C)? Should add the unit of Figure A-D as mg kg-1 dry matter, and also in other place of this manuscript. I thought you measured the concentrations using dried samples.

R.: Done. It would be the aerial part of the plant because the leaves and branches were both analyzed. It was added and the analysis were performed in dried samples.

Line 121-124, please add the source of this sentence, Figure 1?

R.: As explained above, the data for the formula are not presented in any table or figure (data not shown).

Line 143, move (BNI) after the ‘indexes’.

R.: Done.

The full name of AAGR should be unified in Table 3 and Figure 2, and in the text, and change the abbreviation of IC, IU, IV in Figure 2B to GI, UI, and VI.

R.: Done.

Line 139, and Line 142, it should be Figure 2B.

R.: Done.

Line 311, in Table 6, change and Raji[28] Camargo[29] to ‘Raji[28] and Camargo[29]’. In this Table, is the unit is mg kg-1 soil, I suggest authors to add soil to make clearer, and similar line 322.

R.: Done. Just note that in the table 6 we presented the results of the characterization of both soils used in the experiments, in greenhouse and in field, respectively.

Line 327, use abbreviation for Co and Mo in materials and methods, as you have started used them from line 54.

R.: Done.

Line 330, what is the unit of 39%, is it w/v, or w/w?

R.: Done.

I couldn’t clearly understand the meaning of R3, R5, R5.4, could you add more information for these abbreviations? Please move (beginning of grain maturation) in line 341 to line 325, after the ‘R7’.

R.: Done.

Line 347, authors also measured Mo concentration, I suggest authors to add Mo analysis as well in this sentence.

R.: Yes, Mo was analyzed. But Mo was not mentioned because it is a nutrient (micronutrient) and Co is considered as a beneficial element. Due to this, Co analysis was mentioned separately.

Line 379, move ‘(NBI)’ after ‘indexes’.

R.: Done.

Line 280-Line 289, Line 406-Line 413, change the font following the instruction of Plants-Basel.

R.: Done.

Line 452, add (DM) after dry matter.

R.: Done.

Table S2, I suggest authors to separate Co and Mo using a line between parameters of soil (Co) and seed (Mo).

R.: Done.

Reviewer 2 Report

Soybean seed enrichment with cobalt and molybdenum as an

alternative to conventional seed treatment

         In this study, the authors aimed to verify if cobalt and molybdenum application during the reproductive phase of soybean increases seed cobalt and molybdenum concentration without adverse effects on seed quality under greenhouse conditions. The authors investigated foliar or soil application of cobalt and molybdenum and the treatments consisted of cobalt doses combined with molybdenum, and a control without cobalt and molybdenum application. The results showed that the foliar application was more efficient in producing enriched cobalt and molybdenum seed, without adverse effects on nutrition, development, quality, and yield of parent plants and seed. The seed showed higher germination, vigor, and uniformity for the development of soybean seedlings. The authors concluded the application of 20 g per ha Co and 800 g per ha Mo via foliar application at the reproductive stage of soybean increased germination rate and achieve the best growth index and vigor index of enriched seed.

Although the MS is well written and well conducted, there are some critical issues that should be considered to improve the quality of the manuscript. The comments and suggestions are as follows:

-         The chemical and physical analysis of soil is not clear. Please define the soil type and put the percentage of each component for the physical analysis. What about the N content and EC?

-         Line 126-127. The chlorophyll, flavonoid, and N balance indices, germination rate, and soybean yield were not influenced by Co application (Table 3). Why the authors referred only to Co? What about Mo?

-         Table 3. I recommend authors to revise the statistical analysis. For example, in AAGR feature the difference between 20/800 and 30/800 treatments was significant (65 and 68%), however, the difference between control and 10/800 was not  significant (47 and 63%) why? Please revise all means.

-         Table 4. The same comment of Table 3.

-         Line 234-235. Foliar application provided a 24% higher concentration in the seed when compared to soil application. This sentence is not clear. Please re-edit it again. Foliar application of what? provided a 24% higher concentration of what?

-         Line 267-270. In our greenhouse experiment, the seed B concentration increased with increasing Co doses. However, the opposite was observed in the field experiment, where the application of Co and Mo reduced the concentration of B and Mn in the leaves and of Ca, Cu, Mg, Mn, S, and Zn in soybean seeds. This is very interesting result. Please explain why.

-         Line 280-289. Please modify the text format to the correct one as above text.

-         Line 322-323. Why the authors added B, Cu, Fe and Zn to the soil? Is this a practical application used by the producers?

-          Line 331. Foliar application was performed using a 4 cm flat brush and precision balance. Is a brush considered a practical tool for foliar application of microelements?

-         Line 335. The greenhouse experiment was set up in a completely randomized design with a 4x2+1 factorial. However, the field experiment was set up in a randomized complete block design (Line 357). Please explain why the design was changed although the same treatments were applied.

-         Line 347, 390, 425. From the text it appeared that in the soybean herb the authors investigated Co only, however they determined both Co and Mo in soil samples? But in the conclusion (Line 438), they reported that foliar application of Co and Mo at the R5.4 reproductive stage of soybean parent plants increases the concentration of Co and Mo in seed. Please, revise.

-         Lines 406-413. Please modify the text format to the correct one as above text.

Author Response

Responses to Reviewer 2

Dear reviewer,

The authors would like to sincerely thank you for the useful comments and suggestions, which were carefully considered and incorporated into the revised manuscript. The revised manuscript was also improved: 1)  the introduction was improved to provide sufficient background, and relevant references were included, 2) cited references were contextualized to highlight the relevance of the research, 3) the conclusions were improved to be supported by the results.

Although the MS is well written and well conducted, there are some critical issues that should be considered to improve the quality of the manuscript. The comments and suggestions are as follows:

-         The chemical and physical analysis of soil is not clear. Please define the soil type and put the percentage of each component for the physical analysis. What about the N content and EC?

R.: The analysis description was added (item 4.1.1.). The soil used in the greenhouse experiment was a Typic Quartzipisamment and the field experiment was conducted in a Oxisol area. The N content and EC of the soil were not measured, as those variable were not used for soil fertility analysis and fertilizer recommendation of tropical soils.

-         Line 126-127. The chlorophyll, flavonoid, and N balance indices, germination rate, and soybean yield were not influenced by Co application (Table 3). Why the authors referred only to Co? What about Mo?

R.: It was corrected.

-         Table 3. I recommend authors to revise the statistical analysis. For example, in AAGR feature the difference between 20/800 and 30/800 treatments was significant (65 and 68%), however, the difference between control and 10/800 was not  significant (47 and 63%) why? Please revise all means.

R.: In this statistical analysis, with the Dunnett test, we are only comparing the control versus each treatment. So, there was not difference between 20/800 and 30/800 g ha-1 treatments in the AAGR evaluation, as those treatments are not compared when using Dunnett test. There was difference between the control and 30/800 g ha-1. The same is observed to UI evaluation and VI for the 20/800 g ha-1 doses.

-         Table 4. The same comment of Table 3.

R.: In this statistical analysis, with the Dunnett test, we are only comparing the control versus each treatment as explained above.

-         Line 234-235. Foliar application provided a 24% higher concentration in the seed when compared to soil application. This sentence is not clear. Please re-edit it again. Foliar application of what? provided a 24% higher concentration of what?

R.: Done.

-         Line 267-270. In our greenhouse experiment, the seed B concentration increased with increasing Co doses. However, the opposite was observed in the field experiment, where the application of Co and Mo reduced the concentration of B and Mn in the leaves and of Ca, Cu, Mg, Mn, S, and Zn in soybean seeds. This is very interesting result. Please explain why.

R.: This could have happened because of the different experimental conditions of the greenhouse and field, such as the soil and volume of the soil explored by the roots, evapotranspiration, the different phases of the plant when the shoots were sampled.

-         Line 280-289. Please modify the text format to the correct one as above text.

R.: Done.

-         Line 322-323. Why the authors added B, Cu, Fe and Zn to the soil? Is this a practical application used by the producers?

R.: Yes, the application of micronutrients is a common practice in Brazil, for tropical soils with very low fertility.

-          Line 331. Foliar application was performed using a 4 cm flat brush and precision balance. Is a brush considered a practical tool for foliar application of microelements?

R.: This is a procedure realized in greenhouse experiments to avoid solution runoff from the leaf to the soil. In the field experiment, a CO2 pressurized portable knapsack sprayer was used.

-         Line 335. The greenhouse experiment was set up in a completely randomized design with a 4x2+1 factorial. However, the field experiment was set up in a randomized complete block design (Line 357). Please explain why the design was changed although the same treatments were applied.

R.: This occurred because in a greenhouse experiment there is greater control of random variables, which is more difficult to do under field conditions. Therefore, the different statistical designs were chosen even though the treatments were similar.

-         Line 347, 390, 425. From the text it appeared that in the soybean herb the authors investigated Co only, however they determined both Co and Mo in soil samples? But in the conclusion (Line 438), they reported that foliar application of Co and Mo at the R5.4 reproductive stage of soybean parent plants increases the concentration of Co and Mo in seed. Please, revise.

R.: Yes, Mo was analyzed. But Mo was not mentioned in those sentences because is a nutrient (micronutrient) and Co is considered as a beneficial element not as a nutrient. Due to this, Co was mentioned separately.

-         Lines 406-413. Please modify the text format to the correct one as above text.

R.: Done.

Round 2

Reviewer 2 Report

The authors carefully revised the MS and all comments are addressed in current form. Therefore, the MS can be accepted.